# Liquid Biopsy Combined with Multi-Omics Approaches in Diagnosis, Management, and Progression of Diabetic Retinopathy

**DOI:** 10.3390/biomedicines13061306

**Published:** 2025-05-26

**Authors:** Qing Chen, Yi Chen, Kefan Mou, Ming Zhang

**Affiliations:** 1Department of Ophthalmology, West China Hospital, Sichuan University, Chengdu 610041, China; sunnychenqing@foxmail.com (Q.C.); chenyicycycy0723@gmail.com (Y.C.); moukefan@stu.scu.edu.cn (K.M.); 2Department of Ophthalmology and Research Laboratory of Ophthalmology, West China Hospital, Sichuan University, Chengdu 610041, China

**Keywords:** diabetes, diabetic retinopathy, complications, vascular endothelial growth factor, biomarkers, liquid biopsy

## Abstract

Diabetic retinopathy (DR) is a leading cause of vision impairment worldwide, necessitating early detection and personalized treatment strategies. Liquid biopsy, a minimally invasive diagnostic tool, offers significant advantages over traditional methods by enabling analysis of biofluids such as blood, tears, aqueous humor, and vitreous humor. This review highlights the advances in liquid biopsy techniques, sample collection methods and their applications in protein detection and metabolomics analysis for DR. It also explores the key protein biomarkers, including vascular endothelial growth factor (VEGF), inflammatory cytokines, and matrix metalloproteinases (MMPs), and investigates the associations between different biofluids. Metabolomics of liquid biopsy is emphasized for its role in identifying metabolic biomarkers linked to DR pathogenesis, providing new insights into disease mechanisms and personalized interventions. The challenges of liquid biopsy, such as technical limitations and the need for standardization, are also discussed. Advances in computational tools, bioinformatics, and artificial intelligence are supposed to further enhance multi-omics integration, thereby improving precision medicine in DR care. This comprehensive review brings attention to the transformative potential of liquid biopsy in DR diagnosis and management, with implications for broader ophthalmic and systemic diseases.

## 1. Introduction

Diabetic retinopathy (DR) is a significant ocular complication of diabetes, affecting over 100 million people currently living with DR worldwide [1,2,3]. DR is one of the leading causes of blindness and visual impairment. In addition to its health impact, DR imposes a substantial economic cost [4]. In the United States alone, diabetes-related blindness leads to an estimated annual loss of around USD 500 million [5]. Furthermore, patients with DR, especially those in advanced stages of DR such as diabetic macular edema (DME) and proliferative diabetic retinopathy (PDR), face a significant medical expense burden. This is primarily due to the high cost and frequency of administering anti-vascular endothelial growth factor (VEGF) therapies for managing DME [6].

DR is primarily diagnosed through dilated examinations and fundus photography to detect microaneurysms, hemorrhages, and exudates, as well as by using fundus photography and optical coherence tomography (OCT) to identify subclinical macular edema, thereby achieving early disease detection [7]. In addition, optical coherence tomography angiography (OCTA) can non-invasively visualize microvascular changes such as capillary dropout, while fluorescein angiography (FA) can clearly identify capillary nonperfusion in the macular area and help detect peripheral retinal ischemia and peripheral lesions—such as neovascularization—that may not be clinically apparent [8,9,10]. Early screening of patients living with diabetes to detect and manage DR at its initial stages is essential for preventing disease progression and reducing vision loss [11]. Currently, DR screening primarily relies on ophthalmologists to evaluate fundus photographs. However, variability in physician expertise and imaging equipment can result in misdiagnoses or missed cases [11,12]. In recent years, researchers have introduced the concept of diabetic retinal neurodegeneration, viewing it as a preclinical manifestation of DR [13]. Much evidence suggests that retinal neurodegeneration may precede vascular lesions, with progressive retinal thinning and visual dysfunction already present in diabetic patients before the onset of traditional DR [14,15,16,17]. A visual function working group identified 12 potentially relevant diagnosis testing parameters, of which 8 have preliminary data indicating an association with DR, including microperimetry, static automated perimetry, electroretinogram (ERG) oscillatory potentials, flicker ERG, low-luminance visual acuity, contrast sensitivity, and BCVA [18]. Building on these findings, researchers proposed the term “functional diabetic retinopathy (FDR)” to underscore the importance of functional assessments in early DR screening [19]. Although new imaging modalities and the application of artificial intelligence (AI) are playing new roles in DR imaging, significant challenges persist [20].

Thus, liquid biopsy has emerged as a groundbreaking alternative, expanding the diagnostic toolkit to utilize bodily fluids like blood, vitreous fluid, aqueous humor, and tears in the diagnosis, management, and progression of DR [21]. To date, the reliability of liquid biopsy’s ability to continuously monitor tumor dynamics and detect tumor recurrence has been proven, which makes liquid biopsy gain more attention in the other diseases [22]. Studies on the applications of liquid biopsy in DR are also emerging.

Molecular profiling of liquid biopsies from the human eye enables the detection of locally enriched fluids containing biomolecules, including proteins, metabolites, DNA, and RNA, which are derived from highly specialized ocular tissues [21]. The molecular characterization of ocular diseases in living humans could offer the potential to identify novel diagnostic and therapeutic strategies [23]. Leveraging emerging technologies to stratify diabetic patients with different DR risks could facilitate personalized prevention strategies and make targeted interventions for individuals at high risk of developing or progressing to advanced stages of DR [24].

Although current research on the application of liquid biopsy in DR remains in its early stages, this review seeks to address this important gap by systematically synthesizing the available evidence, identifying emerging biomarker candidates, and outlining future directions for clinical translation. In particular, we emphasize the potential of integrating multi-omics approaches with less conventional sample types—such as vitreous humor, aqueous humor, and tears—to provide localized and dynamic insights into the pathophysiology of DR. Moreover, the incorporation of artificial intelligence into liquid biopsy analysis introduces a novel dimension for biomarker discovery and risk stratification. Together, these innovations offer promising avenues for advancing personalized screening, monitoring, and management strategies in DR, and lay the groundwork for future translational research.

## 2. Liquid Biopsy Techniques

### 2.1. Principles of Liquid Biopsy

The principle of liquid biopsy was first introduced as a medical test that includes minimally invasive blood drawing and detection of circulating tumor cells or disseminated tumor cells in cancer patients in 2020 [25]. In recent years, liquid biopsy has become a pivotal tool that analyzes biomarkers in body fluids like blood, urine, or cerebrospinal fluid. Unlike traditional tissue biopsies, which involve surgically removing a piece of tissue, liquid biopsies offer a less invasive, faster, and more convenient way to access fluid sample collection and monitor diseases [26]. Thus, liquid biopsy has the potential to improve early diagnosis, treatment selection, and patient care.

### 2.2. Common Biofluid Sampling

Blood serum is the most commonly applied biofluids while searching for biomarkers of disease [26]. It can be collected through a relatively simple and minimally invasive way. The circulatory system is a vast network of highways connecting all parts of the body, delivering oxygen and nutrients while also picking up waste products and other substances from different tissues and organs, such as ctDNA, CTCs, miRNAs [27]. Plasma diffusion caused by the breakdown of the blood–retinal barrier participates in the exchange of serum and intraocular fluid [28]. Therefore, serum reflects systemic metabolic and inflammatory states associated with DR.

Vitreous fluid is a clear, gel-like substance that fills the space between the lens and the retina in the eye. Vitreous fluid captures locally enriched fluid containing proteins secreted from highly specialized retinal cells and reflects the local changes in the retina. Vitreous fluid can be collected safely through vitrectomy or vitreous aspiration [29].

There are two ways to collect an aqueous humor sample. The first method is aqueous tap, which is generally a safe and quick procedure performed in an outpatient clinic. The second method is collecting aqueous humor at the beginning of intraocular surgeries, such as cataract, glaucoma, corneal, or vitreoretinal surgeries. Wolf et al. had established a workflow regarding the standardized collection of aqueous humor and vitreous fluid for liquid biopsy analysis. It is worth noting that no more than 100 µL of aqueous humor should be collected once to prevent collapse of the anterior chamber [29].

The tear film, the thin layer of body liquid covering the ocular surface, contains thousands of molecules originating from various sources and serving different functions [30]. The collection methods of tear sample vary. Tears can be conducted in outpatient clinic with Schirmer’s strip, a glass microcapillary tube, or with a flush tear collection approach [31,32,33,34]. The flush tear collection was performed by gently instilling approximately 20–60 µL of 0.9% saline solution into the inferior palpebral fold of the participant’s eye. Participants were then instructed to gently close their eyes and rotate or move the eyes to ensure thorough mixing of saline with tears. Subsequently, they tilted their heads slightly to the side, opened their eyes, and tear fluid was carefully collected using a 15 µL capillary tube while strictly avoiding ocular surface contact [21,34,35,36]. In addition, surgical sponges were usually used to collect tears. Surgical sponges are placed on the outer third part of the lower eyelid margin for 5 to 10 min until they are soaked with tears. Then, the fluid was transferred from sponges to tubes by centrifugation [37]. Cellulose acetate filter rods and polyester wicks, though not as popular as the above methods, are also commonly used [38]. Multi-omics research is an advanced approach in biological sciences that integrates multiple layers of molecular data, including genomics, transcriptomics, proteomics, metabolomics, and epigenomics, to achieve a comprehensive understanding of biological systems and disease mechanisms [39]. Currently, multi-omics technology has helped researchers understand the components of tears. Multi-omics technology has also been utilized for diagnosis, prognosis, and monitoring of ocular disease treatments, and contribute to discover new biomarkers and make personalized medicine of ocular disease [40,41].

Therefore, various body fluids can be selected for DR research. And the intraocular fluids are considered better than serum. Table 1 provides a comparison of serum, vitreous fluid, aqueous humor, and tear protein biomarkers in relation to ocular conditions, offering insights into which fluid should be selected for DR biomarker identification and evaluation in different conditions.

## 3. Liquid Biopsy Proteomics in Diabetic Retinopathy

Liquid biopsy proteomics offers a promising solution for DR. Recent studies have discovered and confirmed serum biomarkers that are crucial for the early identification and tracking of DR, although their specificity remains limited due to overlapping involvement in other systemic inflammatory conditions [42,43]. Although the protein content varies across different intraocular fluids, the intraocular liquid biopsies collect local fluid enriched with proteins derived from highly specialized retinal cells, providing valuable insights into the biomarkers of DR and facilitating early diagnosis, disease monitoring, and treatment evaluation. This review provides a comprehensive evaluation of the latest research on the biomarkers for the diagnosis, management, and progression of DR (Figure 1).

### 3.1. Serum Biomarkers

Serum biomarkers reported in DR can be broadly classified into the following categories based on their functional relevance: inflammatory biomarkers, vascular dysfunction biomarkers, neurodegeneration biomarkers, and metabolic dysregulation biomarkers.

Numerous studies indicate that chronic inflammation plays a significant role in the development of DR. Mild inflammation is the main feature of diabetes. The release of inflammatory factors in a high-glucose environment activates various inflammatory signaling pathways that mediate insulin resistance (IR), vascular injury, and other related pathological processes [44]. Cytokines and chemokines, including interleukin-1β (IL-1β), tumor necrosis factor-alpha (TNF-α), interleukin-6 (IL-6), and interleukin-8 (IL-8), have been the focus of extensive research in the context of DR. Furthermore, the serum concentrations of IL-1β, TNF-α, IL-6, and IL-8, exhibited a positive correlation with the presence of DR and were found to be elevated in the serum of patients with PDR relative to non-proliferative diabetic retinopathy (NPDR), which suggests a potential association between cytokines and the severity of DR [45]. Additionally, some inflammation-related biomarkers have also been explored as potential DR biomarkers. CD40 induces pro-inflammatory responses in endothelial and Müller cells and is essential for the development of DR. CD40 expression is upregulated, enhancing CD40-driven expression of pro-inflammatory molecules that promote inflammatory cell recruitment and sustain the inflammatory milieu in DR patients [46,47]. The CD40 ligand–CD40 signaling pathway serves as an upstream regulator of inflammation and angiogenic responses in DR progression and may be involved in maintaining inflammation-driven neovascularization in PDR [48]. A study involving 205 participants indicated that sCD40L can serve as a predictive factor to distinguish DR patients from non-DR patients, while also demonstrating the capacity to predict the severity of DR [48]. Mannose-binding lectin (MBL) is a protein that plays a key role in leukocyte-mediated phagocytosis of microorganisms and activates the lectin pathway, thereby contributing to the pathogenesis of DR [49]. A total of four previous studies have investigated the relationship between serum MBL levels and DR. All the four studies have shown that patients with DR have higher serum MBL levels than patients without DR. Hokazono et al. found that there was a significant correlation between high MBL levels and severe proliferative DR [50,51,52,53].

Cellular adhesion molecules (CAMs) are also related to inflammation and endothelial dysfunction. The dysregulation of CAMs accelerates the adhesion and transmigration of leukocytes, which contributes to inflammation and lead to tissue fibrosis and microvascular complications [54]. Monocyte chemoattractant protein-1 (MCP-1) is a macrophage recruitment chemokine. MCP-1 level increased in DR patients complicated with DME [55]. In addition, the serum levels of intercellular adhesion molecule-1 (ICAM-1) and vascular adhesion molecule-1 (VCAM-1) also show significant differences between DR and diabetes, and gradually increase as DR progresses to PDR [55,56,57,58,59]. Nevertheless, a prospective cohort study demonstrated that the predictive role of ICAM-1 and VCAM-1 in DR was undefined [60].

Conversely, serum levels of anti-inflammatory factors also changed. IL-1 receptor antagonist (IL-1RA) functions by blocking the IL-1 receptor and preventing the activities of IL-1α and IL-1β [61]. The IL-1RA level is lower in the diabetic group compared to the control group. A prospective cohort study indicated that a decrease in IL-1RA might be a plausible early sign of DR [62,63]. Lipoxin A4 (LXA4) is a key mediator of inflammatory processes, with the function of inhibiting leukocyte activation, suppressing free radical generation, reducing the chemotactic responses of polymorphonuclear neutrophils, and activating monocytes and macrophages [64]. LXA4 is a protective anti-inflammatory lipid mediator, but it was reduced in serum and vitreous of DR [65,66].

DR has long been recognized as one of the microvascular complications of diabetes. Anti-neovascularization is also a therapeutic focus for DR [67]. VEGF is a key angiogenic cytokine that has been extensively studied in the ocular angiogenesis and vascular permeability in DR. Serum VEGF has been confirmed to positively correlate with the severity of DR, while the decreased serum irisin, a recently identified myokine, is associated with DR and shows a negative correlation with VEGF [68,69]. The strong effect of angiopoietins on vascular dysfunction suggests their key role in the pathophysiology of DR. Previous studies have identified the potential for treating DR by mediating Ang/Tie2 signaling [70]. The serum level of Ang-2 has been found to be significantly higher, while the Ang-1 level was significantly lower in PDR patients compared to NPDR patients. Furthermore, the Ang-1/Ang-2 ratio was considered to be a more sensitive biomarker [71,72]. Hypoxia-induced factor 1 (HIF-1), as an angiogenic stimulator that is activated in response to tissue hypoxia, was significantly higher in type 2 diabetes patients with DR compared to those without DR [73]. Erythropoietin (EPO) is a systemic angio-genic factor. Elevated serum levels of EPO have been found to correlate positively with the advanced stages of DR, suggesting its involvement in the ischemic and angiogenic processes of DR, especially during the proliferative stage [74].

The vascular dysfunction of DR is also linked to an imbalance between the degradation and synthesis of the basement membrane, leading to the overexpression of its various components [75]. Matrix metalloproteinases (MMPs) play a key role in extracellular matrix protein degradation. MMP activity is tightly regulated by tissue inhibitors of metalloproteinases (TIMPs) [76]. In DR patients, MMP-2, MMP-9, and TIMP-2 are detected with high levels in the serum. And MMP-9 has been considered as important regulators of the pathogenesis and progression of DR [77,78,79]. In addition, high level of MMP-10 was an independent risk factor of DR in patients with type 1 diabetes (T1DM), indicating its participation in the development of microvascular complications in T1DM [80].

Pigment epithelium-derived factor (PEDF) is known for the multifunctional properties that exerts neuroprotective effects through anti-inflammatory and antioxidant effects. PEDF levels were significantly lower in the DR group compared to type 2 diabetes without a retinopathy group [78,81].

Related to metabolic dysregulation, advanced glycation end-products (AGEs) are compounds that formed when glucose reacts with proteins or fats in a non-enzymatic process called glycation [82]. Diabetes can lead to an increase in the formation and accumulation of AGEs in the retina that contribute to the development and progression of DR [83,84]. The intracellular effects of AGEs are mediated through their binding to the receptor for advanced glycation end-products (RAGE), a transmembrane receptor and member of the immunoglobulin family. Hence, RAGE and S100A12, which can bind to the RAGE, are reported to increase in DR [85].

Hemoglobin A1C (HbA1C) has been reported as an indicator of the effectiveness of diabetes management and an independent factor for the development of DR. Overall glycemic control and/or glucose variability could be indicated by HbA1c and are significant factors associated with the onset of DR. Prospective studies have confirmed that elevated plasma HbA1c was a predictive factor for the progression of DR in T1DM [86,87].

### 3.2. Vitreous Humor Biomarkers

Vitreous sampling provides more comprehensive and direct insights by enabling the detection of local biochemical biomarkers. Quantifying the concentration of key proteins and tracking their variations across different disease stages enhance our understanding of DR progression, including PDR and DME [88].

In conditions of hyperglycemia, intraocular pro-inflammatory molecules are increased, promoting the synthesis of inflammatory cytokines and inflammation-related factors that contribute to pathological vascularization and macular edema of DR [89,90,91]. A large body of studies have explored the involvement of inflammatory chemokine in the development of DR. The results are not entirely consistent with those in plasma, with significant increases in TNF-α, IL-2, IL-4, IL-6, and IL-8, and no significant change in IL-1β between DR and non-DR groups [65,92,93,94,95]. Additionally, a study comparing the vitreous cytokine profiles of patients with PDR to those without PDR found that, in addition to the expected inflammatory factors, human vitreous fluid from patients with PDR also showed increases in IL-15 and IL-16 [96]. Additionally, the level of ICAM-1 was significantly increased in the vitreous from patients with PDR compared to non-diabetic control samples [97].

Vitreous concentrations of some inflammation-related factors also showed the same changes as in serum. The sCD40L was significantly elevated in the vitreous of PDR patients compared with control patients who underwent vitrectomy for conditions unrelated to retinal vascular disease in a comparative study [98]. Similarly to the results in serum, vitreous levels of the MCP-1 in patients with PDR were positively correlated with the severity of the PDR [99].

VEGF is the most classic intraocular target for anti-angiogenic therapy and its high level plays an inevitable role in DR [100]. Furthermore, some researchers have compared VEGF concentrations before and after vitrectomy, revealing a significant reduction in VEGF levels after vitrectomy in most patients. These findings suggest that patients with persistently elevated VEGF levels after vitrectomy are at a higher risk of ocular complications, such as neovascular glaucoma. Moreover, the ratio of postoperative residual VEGF to preoperative VEGF concentration may serve as a predictive indicator of bad prognosis [101,102,103,104]. Moreover, a substantial increase in Ang-2 levels has been identified in PDR vitreous samples [105]. Currently, faricimab, a bispecific antibody targeting both angiopoietin-2 and VEGF-A, has demonstrated efficacy in the treatment of DME [106].

HIF-1α levels were significantly elevated in the vitreous fluid of eyes with PDR, which is consistent with the serum results [107]. Platelet-derived growth factor (PDGF) is a potent signaling protein with the ability to stimulate cell growth engaging angiogenesis [108]. PDGF in vitreous was increased and correlated significantly to the severity of PDR, even though no such differences in serum were found [109,110].

In addition to proteolytic degradation of basement membranes and extracellular matrix (ECM) components, MMPs facilitate new vessel growth by removing physical barriers. Moreover, MMPs proteolytically release VEGF from ECM-associated reservoirs, thereby enhancing VEGF bioavailability and activating the VEGF-driven angiogenic switch [111]. The upregulation of MMPs is linked to angiogenesis and progression of PDR. So far, the family members that research has proven to be related to DR are MMP-1, MMP-2, MMP-9, and MMP-14 [104,112].

As a neurodegeneration cytokine, PEDF did not show differential concentrations in vitreous humor [113]. Retinol-binding protein 3 (RBP3, also known as interstitial retinol-binding protein) is a neuroretina-selective protein secreted by photoreceptors. RBP3 plays a crucial role in transporting retinol within photoreceptors and is believed to have potential protective effects on retinal health [114]. RBP3 mitigates the detrimental effects of hyperglycemia on retinal pigment epithelial cells and the angiogenesis, thereby reducing retinal dysfunction and slowing the progression of DR [115]. Vitreous RBP3 concentration in vitreous shows a decreasing trend with the severity of DR, and was negatively correlated with DR severity, vitreous VEGF levels, inflammatory cytokine levels, and the thickness of the photoreceptor layer [110,116,117,118]. These findings indicate that vitreous RBP3 concentration could serve as a critical endogenous retinal protective factor and a potential biomarker for assessing the severity of DR.

### 3.3. Aqueous Humor Biomarkers

Proteomic analysis of aqueous humor (AH) is challenging due to its limited volume and low protein concentration. Recently, the development of aptamer-based assay technology has made it possible to identify proteins in the AH that are difficult to identify by nano-LC-ESI-MS/MS [119,120]. In addition, some studies found that there is fluid flow between aqueous and vitreous compartments [121]. Therefore, the proteins produced and secreted from the posterior segment can be detected in AH [119].

Compared with the healthy control group, cytokines significantly elevated in PDR patients included IL-1β, IL-6, IL-8, TNF-α, and VEGF in the aqueous humor [45,94]. Compared to the eyes of early DR and non-diabetic participants, the level of MCP-1 is elevated in eyes with severe NPDR and PDR. However, more recent studies suggest that there is no difference in MCP-1 level among the non-retinopathy (NDR), NPDR, and PDR groups [122,123,124]. No significant difference was found for other cytokines. Notably, IL-6, IL-8, and VEGF levels in aqueous humor were significantly correlated with those in vitreous fluid [125].

Elevated concentrations of TGF-β and MMP-3 in the AH, but not in the serum, suggest that the intraocular regulation of these cytokines engaged in biological processes independently of systemic regulation. This finding implies a potential role for these cytokines in the progression of DR [126]. The concentration of VEGF can be a biomarker to predict response to anti-VEGF therapy in individuals with high vitreous VEGF levels [127]. Aqueous RBP3 showed a negative correlation with disease severity, following a trend similar to that observed in the vitreous fluid [128].

### 3.4. Tear Biomarkers

Research on tear proteomics is still limited. But existing studies have identified several potential biomarkers for DR. Madania Amorim et al. found that tears of DR patients have higher IL-2, IL-5, IL-18, and TNF concentrations than the heathy controls [129]. Hyun-Jung Kim et al. analyzed the tear proteome and identified different 2-DE maps of tears from DR patients and healthy individuals [130]. Among these proteins, three proteins (LCN-1, HSP27, and B2M) were found to exhibit a progressive reduction in DR groups. Éva Csősz et al. used iTRAQ fourplex labels on a sample to quantitative analyze proteins in the tear fluid and identified significantly higher levels of lipocalin 1, lactotransferrin, lacritin, lysozyme C, lipophilin A, and immunoglobulin lambda chain in the tears of patients with DR [131].

Overall, we observed significant differences in biomarker levels measured between serum and vitreous humor, with Figure 2 highlighting the discrepancies. These variations are primarily driven by several biological factors, including the selective permeability of the blood–retinal barrier, local production of biomarkers within ocular tissues, and differential distribution of systemic biomarkers within the ocular environment [42,132]. These discrepancies do not necessarily impede the prediction and monitoring of DR. On the contrary, they offer important insights [42,43]. Recognizing these discrepancies not only reveals complementary aspects of DR pathogenesis but also enhances the integration of systemic and local biomarkers, providing a more accurate and holistic understanding of disease progression. Thus, these insights significantly enhance the scientific value and translational potential of liquid biopsy approaches in DR.

## 4. Liquid Biopsy Metabolomics in Diabetic Retinopathy

The study of metabolites within biological systems and their changes under different conditions is known as metabolomics [133]. By analyzing metabolites in various biological samples such as blood, urine, and tissues, it is possible for researchers to gain a deeper understanding of the metabolic pathways associated with these diseases. For ocular diseases, metabolomics of intraocular fluids can reveal metabolic processes within the eye, facilitating the identification and treatment of eye diseases. This knowledge allows the identification of specific metabolites linked to various ocular diseases, which can be used as biomarkers in diagnostic analysis [134]. Moreover, ocular metabolomics can uncover individual differences in metabolite profiles and enable the development of personalized treatment plans tailored to specific patients [135]. In 2016, Xuan et al. identified metabolic alterations linked to DR, including changes in amino acid metabolism, lipid metabolism, and energy metabolism [136].

Studies of serum metabolomics identified numerous potential biomarkers, including amino acids, 2,4-dihydroxybutyric acid, leukotrienes, niacin, pyrimidine, purine, arginine, citrulline, glutamic semialdehyde, dehydroxycarnitine, ribonic acid, ribitol, and 3,4-dihydroxybenzoic acid [137,138]. Notably, 3,4-dihydroxybenzoic acid has been identified as an independent risk marker for the progression of DR stages in an untargeted metabolomics study of T1DM patients [137].

Studies based on vitreous metabolomics have shown a broad upregulation of metabolites such as allantoin, α-ketoglutarate, dimethylglycine, lactate, proline, and pyruvate, alongside a decrease in the levels of ascorbic acid, 5-oxoproline, and fumarate. Additionally, glycolysis is downregulated, while the pentose phosphate pathway is activated [139,140].

## 5. Insights into the Clinical Workflow of Liquid Biopsy for Diabetic Retinopathy

To facilitate the clinical translation of liquid biopsy research, it is essential to map key biomarkers across sample types and DR stages. The complexity of DR progression, from NDR to NPDR, PDR, and DME, requires an insightful understanding of how biofluid samples and analytical platforms correlate with diagnostic and prognostic biomarkers. Despite current limitations in sample accessibility, especially for vitreous humor, the concept of liquid biopsy in DR remains highly relevant as a research and translational tool. The clinical pathway for liquid biopsy is still evolving, and its integration into practice will likely be stratified according to disease stage and sample availability. For instance, serum and tear biomarkers may aid in early diagnosis and risk stratification, while vitreous analysis—though invasive—can inform therapeutic decisions during surgery for advanced DR [101,102,103,104]. As such, the value of liquid biopsy lies not in universal applicability at all stages, but in its potential to guide personalized diagnosis and treatment when appropriately applied. Table 2 summarizes the feasible clinical workflows of liquid biopsy in diabetes and DR, including the disease stage, sample type, analytical method, key biomarkers, and clinical value. This integrated overview can serve as a reference for guiding future clinical implementation and research focus. Notably, aqueous humor sampling in NDR patients is not a recommended routine component of DR screening, but it can be opportunistically performed during cataract surgery.

Several biomarkers listed in Table 2 have demonstrated clear or potential predictive value in DR. Among inflammatory biomarkers, TNF-α, IL-6, IL-8, and sCD40L have consistently demonstrated associations with DR severity and progression, making them promising candidates for early diagnosis and disease staging [45,48,62,63,94]. Specially, MCP-1, a potent chemokine involved in monocyte recruitment and activation, has been specifically highlighted for its predictive capability regarding the development of DME and severity of PDR [55,99]. It also shows promise in predicting the progression of PDR following therapeutic interventions, thereby providing clinicians with critical insights to guide personalized treatment strategies and postoperative management [141].

Angiogenic biomarkers such as VEGF, Ang-1, and Ang-2 are well-established predictors of DR severity and therapeutic outcomes. VEGF levels, particularly intraocular VEGF concentrations, serve as robust indicators for assessing disease activity, guiding treatment selection, and predicting the efficacy of anti-VEGF therapies [101,102,103,104,127]. Elevated Ang-2 combined with reduced Ang-1, reflected in the Ang-1/Ang-2 ratio, further refines the prediction of DR progression, underscoring the complex angiogenic signaling mechanisms at play [71,72].

Neuroprotective biomarkers, particularly RBP3, provide additional predictive insights. Reduced levels of RBP3, which is selectively expressed in retinal tissues, correlate with disease severity, retinal degeneration, and progression toward advanced DR stages. Therefore, RBP3 serves as both a predictive biomarker and a potential therapeutic target [128].

In summary, the predictive biomarkers discussed above offer substantial potential to enhance personalized clinical decision-making in DR. Despite promising preliminary evidence, rigorous validation through large-scale prospective clinical studies remains essential to confirm their practical applicability and to establish standardized protocols for clinical integration.

## 6. Advantages, Challenges and Future Perspectives

### 6.1. Advantages of Liquid Biopsy in Diabetic Retinopathy

Liquid biopsy offers numerous advantages in various body fluids. Its primary benefit is non-invasiveness. Unlike other tissues, the eyeball is a non-regenerative organ that makes direct biopsy unfeasible for in vivo analysis. Liquid biopsy, which typically requires only a blood sample and broadens its applicability beyond blood to include vitreous humor, aqueous humor, and tears, allows for timely and frequent monitoring and provides persistent insights into DR for early detection of disease progression and treatment response [23]. Enzyme-linked immunosorbent assay (ELISA) is a widely used and well-established immunoassay technique for the quantification of single proteins, offering high sensitivity and specificity [142]. Moreover, compared to ELISA, aptamer-based assays, which utilize short synthetic nucleic acids for target recognition, offer advantages in stability, multiplexing potential, low sample volume requirement, and cost-effectiveness, making them especially suitable for analyzing scarce samples like aqueous humor or vitreous humor in DR [143].

### 6.2. Challenges and Current Limitations

Despite these advantages, significant challenges persist. While intraocular biopsy techniques, such as vitreous or aqueous humor sampling, are well-established techniques, they are not entirely risk-free. Potential complications of vitreous humor sampling include endophthalmitis, bleeding, and retinal detachment, with an overall low incidence but an increased risk in diabetic patients due to impaired wound healing and a compromised immune response [144,145,146,147,148]. The risk of endophthalmitis is particularly concerning in diabetic patients due to alterations in immune and inflammatory responses involved in wound healing, as well as changes in the bacterial flora of the ocular adnexa [149,150]. Additionally, vitreous biopsy may carry a small risk of inducing vitreous hemorrhage or worsening retinal traction in patients with advanced diabetic retinopathy [151,152]. Aqueous tap is generally considered a low-risk procedure when performed under sterile conditions by experienced clinicians. Evidence has demonstrated the safety of aqueous tap in clinical practice, with serious complications being rarely reported, such as infection, hyphema, lens trauma, or severe inflammation including hypopyon and anterior chamber fibrin formation [153,154]. Currently, there is one established protocol for collecting, annotating, and biobanking aqueous humor and vitreous humor from humans for downstream molecular analyses, including proteomics and metabolomics. Further standardization is required to ensure consistent, reproducible results [29].

Another challenge of liquid biopsy is its high cost, which limits its widespread clinical application. The overall cost of liquid biopsy depends on multiple factors, including the type of biomarker analyzed (e.g., proteins, nucleic acids, metabolites), the detection technology used (e.g., ELISA, next-generation sequencing), and the sample processing requirements [155]. Compared to traditional DR screening methods, such as fundus photography or OCT, liquid biopsy often requires expensive reagents, specialized laboratory equipment, and trained personnel, significantly increasing the financial burden on healthcare systems [156]. Currently, five liquid biopsy diagnostic tests have been approved by the U.S. Food and Drug Administration (FDA) for clinical use, including those for the diagnosis and treatment selection of breast cancer, non-small-cell lung cancer, and colorectal cancer [155]. However, studies related to cost-effectiveness of liquid biopsy in colorectal cancer and non-small-cell lung cancer have indicated that liquid biopsy does not provide a high cost–benefit ratio [157,158]. Nevertheless, technological advancements and miniaturized diagnostic platforms will eventually reduce costs.

### 6.3. Emerging Role of Artificial Intelligence

Looking ahead, with advances in computational tools, bioinformatics platforms, and AI, liquid biopsy shows potential to overcome these barriers [159]. AI has been successfully utilized to automate the detection of retinal diseases from retinal images, particularly in the screening of diabetic retinopathy [160]. In recent years, there has been an explosive growth in advanced technologies for precision medicine, including liquid biopsy and AI for data analysis. The integration of these technologies enables a more comprehensive understanding of disease states, further advancing precision medicine [161]. For example, AI based on liquid biopsy has been applied to assist in bladder cancer diagnosis, while AI has been used to normalize cancer-specific cell-free RNA, enhancing the potential of liquid transcriptomics for diagnostic prediction [162,163]. Recent studies have integrated proteomics from liquid biopsy with single-cell transcriptomics to trace the cellular origins of proteins detected in intraocular fluid, leading to the identification of hundreds of cell-specific protein biomarkers and developed AI models to assess individual cellular aging [119].

By leveraging machine learning and deep learning algorithms, AI can process complex multi-omics datasets, laying the foundation for the routine application of liquid biopsy in DR screening, monitoring, and management. AI models can integrate multi-omics data to uncover novel biomarker interactions and improve predictive modeling for DR progression. Additionally, AI-driven prognostic models stratify DR risk based on biomarker signatures, enabling personalized disease management and early intervention strategies.

### 6.4. Clinical Validation and Future Directions

Current research on diagnostic and prognostic biomarkers for DR based on liquid biopsy primarily remains confined to exploratory clinical trials. Recent clinical studies have demonstrated the predictive value of liquid biopsy biomarkers. Lei et al. reported that elevated aqueous humor MCP-1 levels effectively predict the progression in PDR patients receiving intravitreal bevacizumab combined with vitrectomy [141]. Additionally, another study identified increased vitreous fluid levels of IL-8 as biomarkers correlating with DR severity [113]. Similarly, Muni et al. demonstrated that serum high-sensitivity ICAM-1 is a significant predictor of DR progression [164].

Several clinical trials (e.g., NCT03264976, NCT05079399, NCT05333055, and NCT05944640) continue to explore proteomic, metabolomic, and inflammatory biomarkers across different fluid types to establish personalized biomarker-driven therapeutic approaches. Additional studies are needed to validate their clinical utility, and develop standardized protocols for their integration into routine care. Future research should focus on conducting large-scale, multi-center prospective studies to validate biomarker panels, standardize sampling and analytical protocols, and fully integrate AI models into routine clinical management of DR.

## 7. Conclusions

In summary, liquid biopsy has emerged as a transformative approach for the early detection, prognosis, and personalized management of DR. Through the analysis of key biofluids such as blood, tears, aqueous, and vitreous humor, liquid biopsy offers unique insights into the molecular mechanisms of DR. Advances in proteomics, metabolomics, and even microbiome analysis have significantly expanded the range of potential biomarkers, revealing the complex interactions between systemic metabolic processes, inflammation, and microbial dysbiosis in DR. Integrating AI into the analysis of multi-omics liquid biopsy data holds the potential to enhance the accuracy of clinical decision-making by revealing complex biomarker patterns.

## Figures and Tables

**Figure 1 biomedicines-13-01306-f001:**
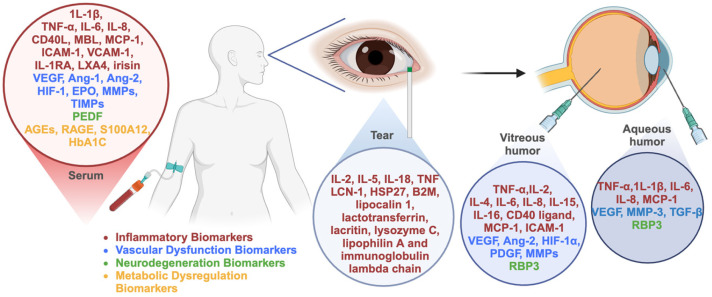
Summary of the biomarkers reviewed in this study.

**Figure 2 biomedicines-13-01306-f002:**
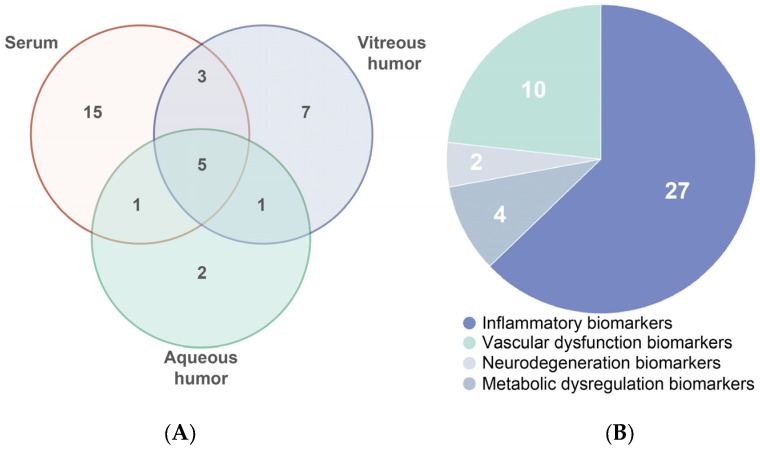
(**A**) Venn diagram showing biomarkers that are common and unique between serum, vitreous humor, and aqueous humor. (**B**) Distribution by function of all 43 biomarkers reviewed in this study.

**Table 1 biomedicines-13-01306-t001:** Common biofluid sampling for ocular biomarkers.

	Serum	Vitreous Fluid	Aqueous Humor	Tears
Composition	Water, proteins, electrolytes, hormones, nutrients	Water, proteins, electrolytes, collagen fibers, hyaluronic acid	Water, proteins, glucose, amino acids, vitamins, electrolytes	Water, electrolytes, lipids, proteins
Collection	Venipuncture	Vitrectomy or vitreous aspiration	Aqueous tap or direct collection at the beginning of intraocular surgeries	Schirmer’s strip or glass microcapillary tube or flush tear collection approach or collection with surgical sponges
Collection site	Outpatient clinic	Operating room	Outpatient clinic or operating room	Outpatient clinic
Protein concentration	6.0–8.3 g/dL	~5 mg/dL	~20 mg/dL	3–4 μg/μL
Invasiveness	Minimally invasive	Highly invasive	Moderately invasive	None
Relevance to ocular disorders	Limited	Significant	Significant	Present

**Table 2 biomedicines-13-01306-t002:** Clinical workflow of liquid biopsy for diabetes and diabetic retinopathy.

Indications	Sample	Analytical Techniques	Key Biomarkers	Clinical Value	Ref.
NDR	Serum	Multiplex Analyses	TNF-α, IL-6, IL-8, sCD40L, MBL, IL-1RA	The early diagnosis of DR	[45,48,50,51,52,53,62,63]
		ELISA	MCP-1	Predicting the risk of DME	[55]
	Tears	ELISA	IL-5, IL-18	The early diagnosis of DR	[129]
		ELISA	HbA1C	Predicting the progression of DR in T1DM	[86,87]
	AH	ELISA	RBP3	Predicting DR progression	[128]
		Multiplex Analyses	TNF-α, IL-1β, IL-6, IL-8, VEGF	DR staging and predicting the risk of PDR	[94]
NPDR	Serum	Multiplex Analyses	IL-1β, VEGF, Ang-1/Ang-2, EPO, MMP-9	Predicting DR progression	[45,68,72,74,77,79]
	VH	Multiplex Analyses	IL-15, IL-16, sCD40L	DR staging and predicting the risk of PDR	[96,98]
	AH	ELISA	TGF-β, MMP-3	Predicting the risk of PDR	[126]
PDR	VH	Multiplex Analyses	MMP-1, MMP-2, MMP-9, MMP-14, RBP3, HIF-1α, MCP-1	Assessing PDR severity	[99,104,107,112]
		ELISA	VEGF, MCP-1	Predicting the progression of PDR after treatment	[101,102,103,104,141]
DME	AH	ELISA	VEGF	Predicting response to anti-VEGF therapy	[127]

## Data Availability

No new data were created or analyzed in this study. Data sharing is not applicable to this article.

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
