# Peer review of "Liquid Biopsy Combined with Multi-Omics Approaches in Diagnosis, Management, and Progression of Diabetic Retinopathy"

_biomedicines, 2025, doi:10.3390/biomedicines13061306_

Round 1

Reviewer 1 Report

Comments and Suggestions for Authors

Thank you for submitting the review article “Liquid Biopsy Combined with Muti-omics Approaches in Diagnosis, Management, and Progression of Diabetic Retinopathy” to MDPI biomedicines. The review provides an in-depth discussion of biomarkers relevant to Diabetic Retinopathy.

In the present form, the review does not surpass the level of listing associations between biomarkers and disease. An adequate evaluation of suitability for DM diagnosis and prediction of disease course as well as embedding into a clinical context is required to increase the level of information in the review article. Therefore, some changes are recommended in the following:

Please lay out the clinical pathway for a liquid biopsy, which includes indication for biopsy, choice of sampling and technique, information of the diagnostic and prognostic value of the chosen biomarker for a given disease-condition (Diabetes, Diabetic Retinopathy).

The authors point towards artificial intelligence to deliver more insight about the usefulness of biomarkers. This section should be expanded including potential strategies on how this can be accomplished.

Please discuss the clinical risks of a biopsy, which include retinal detachment and endophthalmitis (including references). These are generally low but can be increased in diabetic patients. For an indication of a diagnostic biopsy, a collected biomarker has to have some impact/performance that justifies the risk and invasiveness of a biopsy.

There are discrepancies between biomarkers in the blood and vitreous which can lead to challenges in interpretation. Please point out these discrepancies and how they should be interpreted in a diagnostic and predictive context.

L36-38: “Furthermore, the medical expenses for DR patients, particularly those in advanced stages including diabetic macular edema (DME) and proliferative diabetic retinopathy (PDR), are considerably higher than for other diabetes-related conditions.” The sentence is confusing because DR is also a diabetes-related condition.

L127: The following sentence appears lost and is not followed after the introductory sentence: “Serum biomarkers reported in DR can be broadly classified into the following categories based on their functional relevance.”

Comments on the Quality of English Language

There are some errors; improvement recommended. 

Reviewer 2 Report

Comments and Suggestions for Authors

Overview: The authors have provided a review on the liquid biopsy of body fluids – serum, vitreous, aqueous and tear as a potential diagnostic and prognostic biomarkers in DR management. With recent software and analytical platform, it is possible to embrace the technique of liquid biopsy in the DR care. Overall, this is a good article.

The only concern is that biomarkers described are mostly inflammatory markers, and many of them may not be specific to DR. Please address the comments raised in the specific comments.

Specific comments

L2: Title - You have missed the spelling in the title (Muti-omics)!

Abstract

L17: Please correct the sentence (Grammer).

Introduction

L42: We should not be writing “diabetic patients”. It should be “patients living with diabetes”.

Q: You may briefly add the current diagnostic methods for DR in the Introduction, may highlight the early DR.

L43-44: In this context targeting early-stage DR, you may highlight the emerging new concept in DR management. A new term has been coined – functional diabetic retinopathy. Link below:

https://doi.org/10.1016/j.survophthal.2024.11.010

L85, 96: The functions of both the vitreous and tear film are not complete. Either do not mention the incomplete functions or list them fully. You have not mentioned for aqueous, so you may omit the functions in all three fluids.

L101: Write safe or you can delete after the comma as all these techniques are non-invasive.

Plus complete the description of the flush tear collection approach, which is incomplete now.

L107: Briefly describe what is multi-omics.

L111: Correct the first sentence.

L118-119: How correct is your table detail in the last row for serum considering the fact that you mentioned in this sentence? It is contrasting. Please address accordingly.

NB: In general, most of the inflammatory biomarkers are not specific to DR. It is a very complex interaction influenced by many pathological and physiological factors. So can we use them for DR alone and how reliable are they?

L161: You could have abbreviated CAM in L156, where you mentioned CAM for the first time.

L259: Define ECM.

L349: In disadvantages, can you also describe about the cost aspect of the liquid biopsy? How feasible is it to perform in mass of developing countries with poor DR care system?

Comments on the Quality of English Language

Please see specific comments.

Round 2

Reviewer 1 Report

Comments and Suggestions for Authors

Dear Authors,

Thank you for responding to the reviewer comments. Please find further comments and corrections in the following:

The authors write in the corrections to review round 1: “For a diagnostic biopsy to be justified, the collected biomarker must have substantial clinical value in guiding diagnosis, prognosis, or treatment decisions. However, the clinical accessibility of vitreous humor is currently limited, as it is primarily obtained during surgery in patients with PDR. This constraint reduces its applicability for early diagnosis and intervention in diabetic retinopathy.” If there is a reduced applicability for a diagnostic intervention, what is the rationale of the review? It appears contradictory that the authors, on one hand advocate for liquid biopsy and on the other, highlight the limitations. What is the overall judgement? What is the use case scenario?

This comment is linked to Comment 1 of the first review round. For comment 1, the authors addressed the question in their rebuttal but did not make any changes to the manuscript. The question is central to justify the rationale of the manuscript. Please elaborate further in the manuscript and rebuttal letter.

In the response to comment 4, the authors write “….potentially resulting in markedly elevated vitreous levels of certain biomarkers compared to their plasma counterparts.” The response is highly speculative. In addition, the authors did not make any changes to the manuscript. Please avoid any speculative passages in the manuscript. Please modify the manuscript in order to provide new insight and knowledge to the reader to justify the scientific value of the manuscript.

Please avoid over-citation of the same author (Wolf et al.).

In the present form, the manuscript aims to highlight advantages of liquid biopsy but also stipulates that the technique has “all the limitations”. This is as much as the general ophthalmologist or scientist already knows and provides poor justification for the relevance of the article. Please further discuss the current state of the topic: what is going to be the way forward? What works, what does not? What will the clinician do in the near future? Please try to find a justified position.

Round 3

Reviewer 1 Report

Comments and Suggestions for Authors

I would like to thank the authors for improving their manuscript. There are additional points that could be further elaborated:

It may be relevant to broaden the analysis of the clinical trials mentioned in chapter 6.

Specifically, it would be interesting to comment on which biomarkers do or may have a predictive value.

Overall, the article provides a discussion in a pleasant writing style on the liquid biopsy topic. However, the medical problem has not been properly addressed by research which makes the relevance and novelty of the article debatable.

Round 4

Reviewer 1 Report

Comments and Suggestions for Authors

I want to thank the authors for responding to the reviewer’s comments. The nature and purpose of the work remain controversial because the article reports on an unsolved medical problem. A clear and specific direction towards a solution is not available and has not been provided by the authors despite several reviewer requests. The author’s diligence in responding to reviewers’ comments is acknowledged.